# Post-Traumatic Stress Disorder in People with Visual Impairment Compared with the General Population

**DOI:** 10.3390/ijerph19020619

**Published:** 2022-01-06

**Authors:** Tore Bonsaksen, Audun Brunes, Trond Heir

**Affiliations:** 1Department of Health and Nursing Science, Faculty of Social and Health Studies, Inland Norway University of Applied Sciences, 2418 Elverum, Norway; 2Department of Health, Faculty of Health Studies, VID Specialized University, 4306 Sandnes, Norway; 3Section for Trauma, Catastrophes and Forced Migration-Adults and Elderly, Norwegian Centre for Violence and Traumatic Stress Studies, 0484 Oslo, Norway; aubru@ous-hf.no (A.B.); trond.heir@medisin.uio.no (T.H.); 4Institute of Clinical Medicine, Faculty of Medicine, University of Oslo, 0372 Oslo, Norway

**Keywords:** adverse life events, blindness, post-traumatic stress disorder, rehabilitation, visual impairment

## Abstract

Background: People with a visual impairment appear to have an increased risk of experiencing potentially traumatizing life events and possibly also subsequently developing post-traumatic stress disorder (PTSD). This study investigated the point prevalence of PTSD in people with a visual impairment compared with the general population of Norway and examined factors associated with PTSD among people with a visual impairment. Methods: A telephone-based survey was administered to a probability sample of 1216 adults with a visual impairment. Of these, 736 (61% response rate) participated. A probability sample from the general population served as a reference (*n* = 1792, 36% response rate). PTSD was measured with the PTSD Checklist for the DSM-5 (PCL-5), based on the currently most bothersome event reported from the Life Events Checklist for DSM-5 (LEC-5). We used the DSM-5 diagnostic guidelines to categorize participants as fulfilling the PTSD symptom criteria or not. Results: The prevalence of PTSD was higher among people with a visual impairment than in the general population, both for men (9.0% vs. 3.8%) and women (13.9% vs. 8.5%). The prevalence rates of PTSD from the illness or injury that had caused the vision loss (men 3.9%, women 2.2%) accounted for a considerable part of the difference between the populations. For women, PTSD related to sexual assaults also contributed significantly to a higher PTSD prevalence in the visually impaired versus the general population (5.2% vs. 2.2%), while for men there were no other event categories which resulted in significant differences. Among people with a visual impairment, the higher risk of PTSD was associated with lower age, female gender, having acquired the vision loss, and having other impairments in addition to the vision loss. Conclusion: The higher prevalence of PTSD in people with a visual impairment suggests that vulnerability to mental health problems is associated with serious life events. The higher incidence than in the general population is partly due to the illness or injury that had led to the vision loss and partly due to people with vision loss appearing to be more vulnerable through exposure to other types of potentially traumatizing events, such as sexual abuse.

## 1. Introduction

Visual impairment—the partial or complete loss of vision—affects a great number of people worldwide. In 2020, it was estimated that 295 million people lived with a moderate to severe visual impairment, of whom 43 million were blind [1]. In everyday life, vision loss affects the person’s ability to obtain information about their surroundings [2] and thus makes it difficult to predict and avoid potentially dangerous situations. As a result, visual impairments have been shown to increase the risk of potentially traumatic events, such as falls and injuries [3,4]. A recent Norwegian study found that people with a visual impairment had greater exposure to a broad range of serious life events compared with the general population [5]. Substantial differences between the two populations were observed for exposure to fire or explosion, serious accidents, exposure to toxic substance, sexual assaults, war events, life-threatening illness or injury, and severe human suffering.

A higher exposure to potentially traumatic events also suggests the possibility of more mental health problems among people with a visual impairment. In support of this reasoning, previous studies have linked visual impairments to a high prevalence of post-traumatic stress disorder [6], higher risk of depression and anxiety [7,8], and also with burdensome life experiences, such as loneliness [9]. While mental health disorders have been found to be more frequent among young people with a visual impairment compared with their older counterparts [10], studies have also indicated that older adults in this population have a higher prevalence of a range of mental health disorders compared with similarly-aged people in the general population [11]. Thus, the notion that people with a visual impairment are at an elevated risk of mental health problems appears to be valid across age groups.

While the current state of knowledge suggests that people with a visual impairment are more exposed to potentially traumatic events, few studies have examined the prevalence of post-traumatic stress disorder (PTSD). PTSD may follow an exceptionally threatening or horrifying event, where the person experiencing it felt a severe threat of injury or death. Common symptoms of PTSD are re-experiencing the event in the form of flashbacks or nightmares, avoidance of stimuli associated with the event, alterations in cognition and mood, and increased arousal and reactivity [12]. As identified in a recent review [13], the one study that assessed PTSD prevalence specifically in people with a visual impairment was concerned with adolescents in a war conflict area. This study found a lower prevalence of PTSD among those with impaired vision or hearing compared with those without impairments (4.2% versus 11.4%) [14], which was explained by a lower exposure to traumatic events among those with impairments. However, in previous reviews, the prevalence estimates of PTSD in populations prone to visual impairments (older people, primary care patients) have ranged from 1.7% to 32.5% [15,16], which is both higher and lower than those found in general population samples [17,18]. Thus, more research is needed to conclude whether individuals with a visual impairment are at a higher risk of PTSD. Further, research is needed to assess which life events are responsible for the possible difference in the PTSD prevalence between the visually impaired population and the general population.

The aim of this study was to investigate the prevalence of post-traumatic stress disorder (PTSD) in people with a visual impairment compared with the general population of Norway and to examine the factors associated with PTSD among people with a visual impairment.

## 2. Materials and Methods

### 2.1. Sample

#### 2.1.1. People with a Visual Impairment

An anonymous survey was administered to a probability sample of members of the Norwegian Association of the Blind and Partially Sighted. Eligible participants were association members aged 18 years or older, had a diagnosis of a visual impairment, and were able to speak and understand the Norwegian language. Exclusion criteria were an age under 18 years, not being diagnosed with a visual impairment, and the inability to understand Norwegian. Data were collected through structured telephone interviews between January and May 2017. As most members were of old age, age-stratified sampling was used to involve the entire visually impaired population. The study population was divided into four age groups (years 18–35, 36–50, 51–65, ≥66) and an equal number of members across the different age groups were randomly asked to participate. Of the 1216 members we contacted, 736 (61%) opted to participate in the interview. A flowchart of the sample selection is provided elsewhere [19].

#### 2.1.2. General Population

The norm data on baseline characteristics, serious life events, and PTSD in the general population were extracted from the Norwegian Population Study (NorPop) [20,21]. In the NorPop study, a random sample of 5500 adults aged 18 years or older was drawn from the general population. Exclusion criteria were an age under 18 years, not living in Norway, and the inability to understand Norwegian. The data were collected between 2014 and 2015 by self-administered postal questionnaires. Of the eligible participants, 9 people had died, 21 were not able to fill out the questionnaire, and 499 inquiries had non-valid addresses. Of the remaining 4971 inquiries, 1792 individuals (36%) responded by completing and returning the questionnaire, and the sample was deemed to be fairly representative of the Norwegian general population [21].

### 2.2. Measures

#### 2.2.1. Serious Life Events

In both surveys, the lifetime exposure to serious life events was assessed by the Life Events Checklist for DSM-5 (LEC-5). When previously used in a variety of study populations, the LEC-5 has been shown to be reliable and valid [22]. The questionnaire screens for 17 different categories of serious life events (e.g., fire or explosion, traffic accident, serious accident at home, work or during leisure time, physical or sexual assaults, war combat, and life-threatening illness or injury). In the visually impaired population, one additional category was added to the questionnaire: illness or injury causing vision loss. Illness or injuries causing vision loss are unique events in this population and has been shown to induce severe stress reactions [23].

#### 2.2.2. Post-Traumatic Stress Disorder

PTSD symptoms were measured with the PCL-5. This self-administered questionnaire comprises 20 items assessing the full domain of the DSM-5 PTSD diagnosis [24]. The measure has four subscales, corresponding to each of the DSM-5 symptom clusters. Each item was scored on a five-point scale (0, not at all; 1, a little; 2, moderately; 3, quite a bit; 4, extremely) to rate the extent to which the 20 symptoms bothered the study participants during the past month. To categorize participants as fulfilling the PTSD symptom criteria or not, the DSM-5 diagnostic guidelines [12] were applied to the PCL-5. Participants with scores of 2 or above on at least one of the five re-experiencing symptoms, one of the two avoidance symptoms, two of the seven symptoms of negative alterations in cognition and mood, and two of the six arousal symptoms were classified as fulfilling the PTSD symptom criteria [24,25].

#### 2.2.3. Sociodemographic Information

In both surveys, data were collected about the participants’ age, gender, size of place of residence (urban, i.e., >20,000 inhabitants and rural, i.e., ≤20,000 inhabitants), education (years ≤ 10, 11–13, ≥14), employment status (employed or in education versus not), and marital status (married/cohabitant versus not).

#### 2.2.4. Vision-Specific Information and Other Impairments

Among the people with a visual impairment, we included self-reported information about the degree of visual impairment (blindness versus moderate to severe impairment), nature of vision loss (congenital versus acquired), and having other impairments in addition to vision loss (no versus yes). In this study, the self-reported degree of visual impairment refers to the self-reported information that had previously been categorized by an ophthalmologist or other specialist in this field.

### 2.3. Statistical Analyses

All statistical analyses were performed using the SPSS version 26 (IBM Corporation, Armonk, NY, USA) [26]. The significance level was set at *p* < 0.05. We calculated the prevalence of PTSD for men and women in each of the samples separately. The differences in proportions between the samples were examined with Pearson’s chi-square tests if the expected number of PTSD cases in any given cell was 5 or greater; otherwise, we used Fisher’s exact tests. Using a merged dataset containing data from both samples, binary logistic regression analysis was used to examine whether the likelihood of PTSD was different between samples while controlling for sociodemographic covariates (age, sex, education, and marital status). Single and multiple binary logistic regression analyses were also used to examine the factors associated with PTSD in the sample with a visual impairment. Independent variables included ten-year increase in age, sex, education, marital status, severity of vision loss (moderate to severe visual impairment versus blindness), nature of vision loss (congenital versus acquired), and other impairments in addition to vision loss (yes versus no). In the multivariable analysis, the independent variables were entered in one block. Odds ratio (OR) and adjusted odds ratio (AOR) were reported as effect sizes with the corresponding 95% confidence interval (CI) of the OR/AOR.

## 3. Results

### 3.1. Sample Characteristics

The sample included 736 adults with a visual impairment and 1792 adults from the general population. The visual impairment sample had no missing data among the participants, whereas the percentage of missing data from the general population sample ranged between 0% and 2% across the different variables.

Table 1 shows the characteristics of males and females from the sample of people with a visual impairment and the general population. Male and female participants with a visual impairment had lower levels of education, were more often unemployed, and were to a less likely to be married compared with the general population. The mean age of men with a visual impairment was lower than in the general population. Among the women with a visual impairment, the proportion living in urban areas was lower than in the general population.

Among the people with a visual impairment, the onset age of the vision loss ranged from 0 to 76 years (mean: 19 years) and was primarily caused by diseases (50%), followed by congenital causes (43%), and injuries (7%). In total, 25% had self-reported blindness and the remaining 75% had a self-reported moderate to severe impairment, and 35% reported other impairments in addition to their vision loss.

### 3.2. PTSD

The prevalence rates in men (9.0%, 95% CI: 5.9–12.1%) and women (13.9%, 95% CI: 10.5–17.3%) with a visual impairment were higher than in the general population of men (3.8%, 95% CI: 2.5–5.1%) and women (8.5%, 95% CI: 6.7–10.3%). The rates were still higher after adjustments for age, education, and marital status, both for men (AOR: 2.30, 95% CI: 1.34–3.94) and women (AOR: 1.56, 95% CI: 1.06–2.28).

The results for event categories causing PTSD in the visual impairment sample are displayed in Table 2. The most common cause of PTSD in both men and women were injury or illness causing vision loss and, additionally for women, physical and sexual assaults. Compared with the general population, women had higher prevalence rates of PTSD due to sexual assaults, captivity, and severe human suffering. Men had no specific event category, other than injury or illness causing vision loss, that resulted in significantly higher PTSD rates than in the general population.

### 3.3. Factors Associated with PTSD

The results from the analysis of the factors associated with PTSD within the visual impairment sample are displayed in Table 3. Adjusting for all variables, PTSD was associated with lower age (AOR: 0.69, 95% CI: 0.60–0.80), female gender (AOR: 1.81, 95% CI: 1.10–2.96), having acquired the vision loss during the course of life (AOR: 2.61, 95% CI: 1.54–4.42), and having other impairments in addition to the vision loss (AOR: 2.57, 95% CI: 1.59–4.16). Compared with those with the highest level of education, those with 11–13 years of education had a lower risk of PTSD (AOR: 0.58, 95% CI: 0.33–1.00), while the risk of PTSD was similar between those with the lowest and the highest levels of education.

## 4. Discussion

### 4.1. Summary of Results

In this cross-sectional study, we found that both men and women with a visual impairment were more likely to have PTSD compared with the general population. Among men, the event most often causing PTSD was the illness or injury that caused the vision loss. Among women, sexual assault was the most frequent event category that caused PTSD. The women had PTSD caused by sexual assault, captivity, and severe human suffering more often than women in the general population. A lower age, being female, acquiring the vision loss, and having other impairments in addition to the vision loss were associated with a higher risk of PTSD.

### 4.2. Prevalence of PTSD

When controlling for the factors known to be associated with PTSD [20,27,28], both men and women with a visual impairment had substantially higher odds of PTSD compared with men and women in the general population. This finding appears to fill a gap in the literature, as previous studies have found high prevalence rates of PTSD in people with a visual impairment [6] but have been unable to conclude whether they have a higher risk of PTSD compared with the general population [13].

Among men, the illness or injury that caused the vision loss was the most frequent event that caused PTSD. Among women, several event categories caused PTSD more often than in the general population, with sexual assault being the most frequent. Thus, the higher incidence of PTSD in the visually impaired population is partly due to the illness or injury that led to the vision loss and partly due to the people with vision loss appearing to be more vulnerable through exposure to other types of potentially traumatizing events. Previous research has indicated that adverse events, such as bullying and abuse, are commonly experienced among people with a visual impairment—such experiences have been reported among 20–30% of the visually impaired population—and reporting such experiences has been associated with higher levels of depression [29]. People with a visual impairment are also more likely to report discrimination in general [30]. Moreover, sexual abuse has been found to be more prevalent among women with a visual impairment than in the general population (17.4% versus 10.0%) [31]. Thus, in comparison with the general population, the higher levels of PTSD among people with a visual impairment may be a direct result of their higher exposure to potentially traumatizing events.

However, it is also possible that exposure to adverse events may elicit stronger reactions among those with a visual impairment. Due to their limited access to information, people with a visual impairment may have difficulty creating a mental image of the event or knowing their role in the event. This can hinder them from processing or overcoming what they have experienced and thus, create stronger stress reactions than if they could easily process the event [32]. Vision loss may also decrease the person’s ability to respond appropriately in the face of adverse events of any kind, be they natural disasters, traffic accidents or assaults. Having lower self-efficacy under stressful circumstances may increase the susceptibility to experiencing more stress reactions, as previously demonstrated in different populations [33,34].

To the extent that a resource such as education serves to decrease the risk of PTSD among those exposed to trauma, people with a visual impairment appear to have less access to this resource compared with the general population (Table 1). In turn, this may add to the PTSD burden among those with a visual impairment.

### 4.3. Factors Associated with PTSD

A lower age and female gender were associated with a higher risk of PTSD. The higher risk of PTSD in women with a visual impairment is identical to the pattern found in general population samples [20,35,36] and, as found in this study, almost half of the women with PTSD had the disorder caused by having experienced sexual assault. We also found that a younger age was associated with PTSD. This association was not found to be statistically significant in the previous study of the Norwegian general population [20], whereas it has been significantly associated with PTSD in several studies investigating post-traumatic stress reactions in the population during the early stage of the COVID-19 outbreak [37,38,39]. Thus, sociodemographic covariates to PTSD among people with a visual impairment appear to largely mirror those found in the general population.

Having acquired the vision loss and having other impairments in addition to the vision loss were also associated with a higher risk of PTSD. For adults who lost their vision early in life, the condition may be considered a part of who they are rather than something that has happened to them. Conversely, acquired vision loss caused by illness or injury may represent a severe trauma and existential crisis [6], and many of those with PTSD referred to the illness or injury causing the vision loss as the event causing PTSD. Having additional impairments was also associated with a higher risk of PTSD. Having other impairments, such as movement impairments, hearing loss, and cognitive limitations, may create additional barriers to mobility and human movement compared with the vision loss alone or decrease the person’s ability to predict and prevent accidents and other serious life events [3,4]. Multiple impairments may also increase the vulnerability to stress reactions and adverse mental health in general [29].

### 4.4. Strengths and Limitations

The relatively large probability sample of adults with a visual impairment and the use of validated instruments in the assessment of serious life events and PTSD are strengths of the study. Moreover, the ability to compare estimates for people with a visual impairment with estimates from a representative general population sample increases the value of the study.

The limitations of the study include the representativeness of the population of people with a visual impairment. The sample was recruited via a member organization for the blind and partially sighted. Compared with census data from Statistics Norway [40], gender, employment, and place of residence did not differ for our study participants, but their level of education was higher. The rates of people with self-rated blindness were also higher than reported previously [41]. In addition, the use of self-reports on serious life events may have affected the validity of the results. For example, recall bias is a common problem related to the retrospective reports of serious life events. Some events may be forgotten or no longer considered important, whereas others may be amplified in the participant’s memory [42].

While the PCL-5 is a questionnaire designed to be completed in writing, the gold standard for establishing a PTSD diagnosis is the Structured Clinical Interview according to DSM-V (SCID) [43]. However, a previous study found that results from the Norwegian version of the PCL-5 corresponded well with the results from SCID interviews, and the PCL-5 is therefore considered a good measure for assessing PTSD in the general population [44]. In our study, the PCL-5 was used as an interview, where the participants had questions and response options read out loud to them. Therefore, in our opinion, using the PCL in the form of an interview serves as a combination of the two methods and the results are likely to be valid and reliable.

Lastly, non-participation may have caused biased prevalence estimates in both study samples. The available information about non-participants is limited and we do not know how non-responding might have affected the results.

### 4.5. Implications

The high prevalence of PTSD in people with a visual impairment has important implications for prevention and treatment. Mental health adversities due to high levels of exposure to serious life events calls for the facilitation of security in the physical environment [5]. Emphasis should be placed on universal design, and safety and ease of use for people with vision loss. Adaptations must apply to housing, schools, workplaces, leisure activities, the transport sector, and the public sphere.

The high PTSD rates due to the physical and sexual abuse of women suggest a need for preventive measures as well as professional assistance when such events occur. Strategies for prevention include raising public awareness and professional knowledge about the vulnerability of specific groups [45]. The stress reactions caused by a loss of vision appear to have a major impact on mental health. The loss of vision can occur suddenly or develop gradually, with stress caused by not knowing how the vision loss will progress. The risk of mental health problems, such as PTSD, in people who lose their sight must be recognized and met with information about common mental reactions and an offer of follow-up by professionals.

We have previously demonstrated the higher levels of depression in the same study population [10,29] as well as an unmet need for mental health services [10]. The high rates of PTSD support the proposal for a better adapted health care system for people with a visual impairment. People with a visual impairment may have a higher threshold for seeking help. Equally important, there is a lack of knowledge among health professionals about the mental adversities associated with visual impairments [46]. Norway is one of very few countries that is in the process of meeting this need by establishing a national competence center for visual impairments and mental health. Experiences from this work will be disseminated to the research field.

## 5. Conclusions

Even when adjusting for sociodemographic covariates, the prevalence of PTSD was higher in people with a visual impairment compared with the general population. More exposure to illness or injury causing vision loss and to adverse events, such as sexual assault, appear to be responsible for the higher prevalence of PTSD among people with a visual impairment. PTSD was more frequent among those with acquired vision loss and those with other impairments in addition to the vision loss. Implications for prevention and treatment have been discussed.

## Figures and Tables

**Table 1 ijerph-19-00619-t001:** Sociodemographic characteristics of the participants (VI *n* = 736 and GP *n* = 1792).

	Visually Impaired Population		General Population	
Variables	Men	Women	Men	Women
	N = 333	N = 403	N = 834	N = 945
Age (years), mean (*SD*)	51.1 (17.0) ***	51.7 (17.3)	55.7 (15.9)	51.0 (17.0)
Age (years), range	18–95	18–84	18–93	18–94
*Education, n (%)*				
More than 12 years	163 (48.9) **	172 (42.7) ***	432 (52.0)	517 (54.9)
10–12 years	124 (37.2)	162 (40.2)	336 (45.0)	346 (36.7)
9 years or less	46 (13.8)	69 (17.1)	62 (7.5)	79 (8.4)
*Employment, n (%)*				
Employed or in education	181 (54.4) ***	260 (64.5) ***	526 (63.1)	640 (67.7)
Not employed or in education	152 (45.6)	143 (35.5)	304 (36.5)	299 (31.6)
*Size of place of residence, n (%)*				
1–19,999	172 (51.7)	227 (56.3) **	399 (48.4)	444 (47.3)
20,000 and more	161 (48.3)	176 (43.7)	426 (51.6)	494 (52.7)
*Marital status, n (%)*				
Married or cohabitant	166 (49.8) ***	181 (44.9) ***	634 (76.3)	647 (68.9)

Note. VI is visual impairment, GP is general population. Statistical tests were chi square (categorical variables) and independent *t*-test (age). Differences denoted as statistically significant indicate differences between populations and gender. ** *p* < 0.01, *** *p* < 0.001.

**Table 2 ijerph-19-00619-t002:** Current PTSD caused by serious life events (VI *n* = 736 and GP *n* = 1792).

	Visually Impaired Population		General Population	
Most Serious Life Event	Men (N = 333) *n* (%)	Women (N = 403) *n* (%)	Men (N = 834) *n* (%)	Women (N = 945) *n* (%)
Natural disaster (for example, flood, hurricane, tornado, earthquake)	0	0	1 (0.1)	0
Fire or explosion	1 (0.3)	0	1 (0.1)	4 (0.4)
Transportation accident (for example, car accident, boat accident, train wreck, plane crash)	2 (0.6)	1 (0.2)	5 (0.6)	6 (0.6)
Serious accident at work, home or during recreational activity	2 (0.6)	3 (0.7)	3 (0.4)	6 (0.6)
Exposure to toxic substance (for example, dangerous chemicals, radiation)	0	0	0	0
Physical assault (for example, being attacked, hit, slapped, kicked, beaten up)	3 (0.9)	9 (2.2)	8 (1.0)	10 (1.1)
Assault with a weapon (for example, being shot, stabbed, threatened with a knife, gun, bomb)	1 (0.3)	2 (0.5)	0	8 (0.8)
Sexual assault (rape, attempted rape, made to perform any type of sexual act through force or threat of harm)	1 (0.3)	21 (5.2) **	2 (0.2)	21 (2.2)
Other unwanted or uncomfortable sexual experience	0	0	0	0
Combat or exposure to a war-zone (in the military or as a civilian)	1 (0.3)	0	2 (0.2)	1 (0.1)
Captivity (for example, being kidnapped, abducted, held hostage, prisoner of war)	1 (0.3)	3 (0.7) *	0	0
Life-threatening illness or injury	2 (0.6)	6 (1.5)	3 (0.4)	11 (1.2)
Severe human suffering	2 (0.6)	6 (1.5) ***	0	0
Sudden violent death (for example, homicide, suicide)	1 (0.3)	1 (0.2)	6 (0.7)	7 (0.7)
Sudden accidental death	0	1 (0.2)	1 (0.1)	6 (0.6)
Serious injury, harm or death you caused to someone else	0	0	0	0
Any other very stressful event or experience	0	0	0	0
Illness or injury that caused loss of vision #	13 (3.9)	9 (2.2)	-	-
Total	30 (9.0) ***	56 (13.9) **	32 (3.8)	80 (8.5)

Note. VI is visual impairment, GP is general population. # The item was only used with the VI sample. For events causing current PTSD, there was a maximum of one event per individual overall. Statistically significant differences indicate that for the relevant event and for each gender, the rates of PTSD differed between populations (Fisher’s exact test). * *p* < 0.05, ** *p* < 0.01, *** *p* < 0.001.

**Table 3 ijerph-19-00619-t003:** Factors associated with current PTSD in the VI sample (*n* = 736), unadjusted and adjusted analyses.

	Unadjusted			Adjusted		
	OR	95% CI	*p*	AOR	95% CI	*p*
Age, increase in 10 years	0.77	0.68–0.89	<0.001	0.69	0.60–0.80	<0.001
Females vs. males	1.63	1.02–2.61	<0.05	1.81	1.10–2.96	<0.05
Education						
More than 13 years	Reference			Reference		
11–13 years	0.81	0.49–1.34	0.41	0.58	0.33–1.00	<0.05
10 years or less	1.16	0.62–2.16	0.64	1.11	0.57–2.17	0.76
Married/cohabitant, yes vs. no	0.63	0.40–1.00	0.05	0.64	0.39–1.04	0.07
Blind vs. moderate/severe VI	1.02	0.61–1.71	0.94	1.18	0.68–2.04	0.55
Acquired vs. congenital VI	1.90	1.17–3.07	<0.01	2.61	1.54–4.42	<0.001
Having other impairments	2.51	1.59–3.95	<0.001	2.57	1.59–4.16	<0.001

Note. VI is visual impairment.

## Data Availability

Data are from the research project European Network for Psychosocial Crisis Management–Assisting Disabled in Case of Disaster (EUNAD). Public availability of data may compromise the privacy of the participants. According to the informed consent provided by the participants, the data are to be stored properly and in line with EU Regulation 2017/679 (General Data Protection Regulation [GDPR]). However, anonymized data is available to researchers who provide a methodologically sound proposal in accordance with the informed consent of the participants. Interested researchers can contact project leader Trond Heir (trond.heir@medisin.uio.no) with a request for our study data.

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
