# Peer review of "Post-Traumatic Stress Disorder in People with Visual Impairment Compared with the General Population"

_ijerph, 2022, doi:10.3390/ijerph19020619_

Round 1

Reviewer 1 Report

Dear Authors,

I find this research to be interesting and valuable for the public. It addresses an important public health matter and gives results that can expand our knowledge and it provides guidelines for further research but also for the public health strategies and interventions.

Here are some suggestions for minor corrections:

In the Methods section:

- excluding criteria is missing for both samples

 - terms rural and urban have wrong "<" and ">" signs

- please explain self-reported information about the degree of visual impairment; was this participants' perception of the degree of the impairment or they self-reported impairment that was earlier categorized by some sort of medical or other specialist or an institution?

  • please clarify the term "ten-year increase in age"

Results section:

-at page 5, line 173-175, please clarify that this refers to VI population. Please comment the most common causes for PTSD in your general population sample.

Discussion section:

- lines 249-251 (reference 34) is redundant

- conclusion in lines 251-252 are uncorroborated, please add references concerning other investigated sociodemographic factors that were explored by other authors and compare to your results, then make a conclusion

Author Response

Please see attachement.

Reviewer 2 Report

Dear authors
The introduction presents good support that justifies the proposed research, with precision and clarity. There is little information about it. In particular, concerning the fact that visual impairment can become a risk factor for physical and psychological integrity where PTSD is one of the possible consequences. It is understood that they follow the line of Van der Ham et al., 2021 with a good contribution. However, it is convenient to include some key concepts of PTSD as well as those DSM 5 criteria considered more evident in the populations studied.
In material and methods, it is convenient to specify and justify:
a) The error and reliability data of the sample evaluated in both populations.
b) The LEC 5 format used for both groups evaluated and justified whether both formats used, questionnaire for normative population and interview for the target population, is equivalent in homology to be contrasted.
c) In this sense, it is suggested to incorporate statistical indicators to justify whether or not it corresponds to parametric analysis.
d) Likewise, if the results of the surveys and quantification of the interviews present data reliability indices for both populations.

In the results, the statistical indicators obtained can be indicated, estimating criteria of reliability and homogeneity of the data, to ratify the finding.
Likewise, given the influence of independent variables that influence PTSD, the predictability of this diagnosis could be demonstrated. For example, if the correlations between the independent variables and the total result of the evaluation with LEC 5 were statistically significant, a linear regression model could be used to determine a predictive nature of these variables against PTSD, a relevant aspect for planning future interventions.
In discussion:
With the aforementioned, we could justify what was indicated in line 200 of this discussion and 12 to 314 of Conclusions. In addition to being able to analyze the risk factor involved (line 212) and some proposals for future psychoeducational actions (lines 239, 285, and 286) that are hinted at in the text of the discussion. In addition to showing the strength expressed in lines 266 and 267, and reducing the weakness expressed in lines 276.
What is indicated in lines 293 to 295, exceed what is reported in this paper; It is advisable to reformulate these statements based on the evidence verified with a bibliographic citation and the findings of this manuscript.
The conclusion expresses the main findings concisely. They could be highlighted. Furthermore, if the predictive validity could be examined with the regression model, they could present a greater significance for future actions at a therapeutic and preventive level.

Reviewer 3 Report

'2. Materials and Methods' section

Under '2.2. Measures', it is not clear how PTSD was measured. What questionnaire and what scoring was used?

Under '2.3. statistical analyses', authors mentioned using binary logistic regression analysis, controlling for sociodemographic covariates. If this is the case, they need to report Adjusted Odds Ratios (AORs) not Odds Ratios (ORs). In the Table 3, authors made distinction between unadjusted and adjusted ORs, but consistency and clarification is needed.

'3. Results' section

In the results section, all ORs need to change to AORs, as authors pointed using logistic regressions controlling for sociodemographic covariates.

There are ORs and CIs (i.e., men (OR: 2.30, 95% CI [1.34-3.94]) and women (OR: 1.56, 95% CI [1.06-2.28])) under '3.2. PTSD' that are not presented or referred to any table. It seems they are also AORs not ORs.
